**Data Availability Statement:** All relevant data are within the paper and its Supporting Information files.

# Binary cutpoint and the combined effect of systolic and diastolic blood pressure on cardiovascular disease mortality: A community-based cohort study

Ju-Yeun Lee[1,2,3], Ji Hoon Hong[4,5], Sangjun Lee🔘[1,6,7], Seokyung An🔘[1,6,7], Aesun Shin[1,2,6], Sue K. Park🔘[1,2,6]*

1 Department of Preventive Medicine, Seoul National University College of Medicine, Seoul, Korea, 2 Integrated Major in Innovative Medical Science, Seoul National University College of Medicine, Seoul, South Korea, 3 Department of Ophthalmology, Myongji Hospital, Hanyang University College of Medicine, Goyang, South Korea, 4 Department of Medicine, Seoul National University College of Medicine, Seoul, Korea, 5 Department of Internal Medicine, Seoul National University Hospital, College of Medicine, Seoul, South Korea, 6 Cancer Research Institute, Seoul National University, Seoul, Korea, 7 Department of Biomedical Sciences, Seoul National University Graduate School, Seoul, Republic of Korea

* suepark@snu.ac.kr

## Abstract

### Objectives

This study aimed to examine the risk of cardiovascular disease (CVD) death according to blood pressure levels and systolic and/or diastolic hypertension.

### Methods

From 20,636 cohort participants, 14,375 patients were enrolled after patients with prior hypertension on antihypertensive drugs were excluded. For the combination analysis, participants were divided into four groups (systolic/diastolic hypertension, systolic hypertension only, diastolic hypertension only, and non-hypertension). The risk of CV death was calculated using the hazard ratio (HR) and 95% confidence intervals (95% CI) in a Cox regression model.

### Results

The risk of CVD death increased in systolic hypertension (HR = 1.59, 95% CI 1.26–2.00) and systolic/diastolic hypertension (HR = 1.84, 95% CI 1.51–2.25). The highest risks of hemorrhagic and ischemic stroke were observed in the diastolic hypertension (HR = 4.11, 95% CI 1.40–12.06) and systolic/diastolic hypertension groups (HR = 2.59, 95% CI 1.92–3.50), respectively. The risk of CVD death was drastically increased in those with SBP≥120 mmHg/DBP≥80 mmHg. The highest risk was observed in those with SBP of 130–131 mmHg and 134–137 mmHg.

### Conclusion

The combined analysis of systolic and/or diastolic hypertension appears to be a good predictor of CVD death. The risk of CVD death in the prehypertensive group could be carefully

**Funding:** (A grant from the Korea Health Technology R&D Project through the Korea Health Industry Development Institute (KHIDI), funded by the Ministry of Health & Welfare, Republic of Korea (grant number: HI16C1127)). The funders had no role in study design, data collection and analysis, decision to publish, or preparation of the manuscript.

**Competing interests:** The authors have declared that no competing interests exist.

monitored as well as in the hypertensive group, presumably due to less attention and the lack of antihypertensive treatment.

## Introduction

Cardiovascular disease (CVD) is one of the leading causes of death and disability worldwide [1]. Although a large proportion of CVD is preventable, its prevalence continues to rise [2]. In Korea, CVD is the second leading cause of death and has become a major social burden from a public health perspective [3]. Among the possible risk factors, elevated blood pressure (BP) highly affects the morbidity and mortality of CVDs. It was reported that county-level hypertension-related CVD mortality increased from 362.1 per 100,000 in 2000 to 430.1 per 100,000 in 2019 among adults aged ≥65 years. Elevated BP-related CVD mortality during 2010 to 2019 was found to increase 86.2% among patients aged 35 to 64 years, and 66.1% for patients aged ≥65 years [4].

A significant relationship between various BP components and cardiovascular death was previously reported [5–7]. In a Taiwanese study, CVD mortality risks increased when systolic BP (SBP) was higher than 160 mmHg or diastolic BP (DBP) was higher than 90 mmHg in older adults [8]. In a Korean population, Kim et al. reported a J-curve pattern for DBP on CVD mortality [9]. A meta-analysis reported that primary preventive BP lowering is correlated with a reduced risk of death and CVD when baseline SBP is 140 mmHg or higher [10]. Although the independent risk of either SBP or DBP on the risk of CVD mortality has been demonstrated in previous studies [11], the combined effect of SBP and DBP on CVD mortality has rarely been reported. Moreover, there is still insufficient information regarding a BP cut-off value that could be used in the prevention of CVD mortality. Hence, it is important to collect detailed data on the risks for CVD mortality according to each value of SBP and DBP to help identify the optimal BP cut-off point for predicting CVD mortality.

This study aimed to identify the longitudinal trend of the increase in the risk of cardiovascular death according to SBP and DBP levels and observe specific patterns associated with the increase in CVD mortality risk using a combined analysis of systolic and/or diastolic hypertension. We hypothesize that a specific combination of systolic and diastolic hypertension will more significantly influence the risk of CVD death, which will depend on the specific type of CVD. To prove the above, we conducted a composite analysis of hypertension rather than an independent analysis of systolic and diastolic hypertension.

## Methods

### Data sources and study population

All eligible participants for our study were selected from a prospective community-based cohort study, the Korean Multi-center Cancer Cohort (KMCC) study. All participants in the cohort study were enrolled during 1993 to 2004, and death was followed-up until Dec 31st, 2018. The mean follow-up period was 15 years. Detailed information on the KMCC study is described in a previous article [12]. For this study, we excluded participants who died during the first two years of follow-up (n = 386), those with no age data (n = 1), those with no SBP/DBP data (n = 5,007), and those previously diagnosed with CVD or prior hypertension that were prescribed antihypertensive drugs (n = 867). This study included 14,375 participants, of whom 570 died of CVD (S1 Fig).

Information on age, sex, reproductive factors, occupational and environmental exposures, lifestyle factors including sleep duration, smoking, alcohol use, physical activity, and dietary factors were obtained through structured questionnaire interviews. Anthropometric measurements were obtained, such as weight, height, body mass index, and body circumference. Laboratory data were collected from blood and spot urine samples. This study was conducted in accordance with the tenets of the Declaration of Helsinki, and all procedures involving human participants were approved by the Institutional Review Board (IRB) of Seoul National University Hospital (IRB NO: 1701-021-821). Informed consent was obtained from all participants.

## Classification of SBP and DBP and hypertension definition

Trained staff measured SBP and DBP twice in the right arm using a standardized mercury or automatic sphygmomanometer depending on the institution. Participants were seated for at least 5 min before the BP measurements, and the mean of the two BP readings was recorded for data analysis [13].

For combined analysis of systolic and/or diastolic hypertension on the risk of CVD death and to establish a uniform definition of systolic and diastolic hypertension in this study, several standard values were primarily set, including 'SBP 140 mmHg' and' DBP 90 mmHg' as the hypertension criterion in the traditional guideline, and 'SBP 130 mmHg or 135 mmHg' and 'DBP 85 mmHg or 80 mmHg' as the hypertension criterion based on the pilot spline analysis. Participants were divided into four groups: systolic/diastolic hypertension, systolic hypertension, diastolic hypertension, and non-hypertension for the combined analysis of systolic and/ or diastolic hypertension.

For trend analysis of the risk of CVD death according to BP levels, SBP and DBP levels were divided into high and low BP groups at a specific BP level. For example, at the cutpoint of SBP 120 mmHg, cohort participants were divided into two groups, 'SBP $\geq$ 120 mmHg' and 'SBP $<$ 120 mmHg'.

## Follow-up for main outcomes: Death

Information on the date and cause of death was obtained from the death certificate database of the Korea National Statistics Office. The causes of death were classified according to the International Statistical Classification of Diseases and Related Health Problems 10th Revision (ICD-10) as follows: all-cause death (A00-Z99), all-cause death (I00-I99), ischemic heart disease death (IHD, I20-I25), acute myocardial infarction death (AMI, I21), stroke death (I60-I69), cerebral hemorrhage death (I60-I62), cerebral infarction death (I63), other stroke death (I64-I69), hypertensive diseases death (I10-I15), all cancer death (C00-C97), and non-disease death (S00-T98 or V09-Y98) (S1 Table). Other strokes are defined as occlusion and stenosis of the precerebral and cerebral arteries or other cerebrovascular diseases not specified as cerebral hemorrhage or infarction. Causes of non-disease death include injury, poisoning, the consequence of external factors (ICD-10 code S00-T98), and external factors of morbidity and mortality (ICD-10 code V01-Y98). The total number of deaths was analyzed in four categories: all-cause, CVD-specific cause, cancer-specific cause, and other causes. The death results from cancer-specific causes or non-disease causes are used as a reference data to compare our findings.

## Statistical analysis

The chi-square test for categorical variables and the analysis of variance (ANOVA) test for continuous variables were used to compare the baseline characteristics between groups. The association between each SBP, DBP, and combined BP group and the risks for each category of

death were analyzed using a Cox proportional hazards regression analysis. Hazard ratios (HRs) and the corresponding 95% confidence intervals (CIs) of risk factors were obtained based on the regression coefficients and standard errors from Cox's proportional hazards regression models with follow-up time as the time scale. To estimate the dose-response effect, testing for trends was performed. The HRs for CVD mortality according to each binary cut-off point of SBP (111–151 mmHg) and DBP (70–111 mmHg) by 1 mmHg were also calculated. A proportional hazard assumption test was performed.

Sub-analyses were performed to examine the association between each combined BP group and the risk of CVD mortality according to age, total cholesterol, body mass index (BMI), the presence of anemia, chronic kidney disease, or diabetes. All models were adjusted for age, sex, past medical history of diabetes, family history of CVD, BMI ($<25$ or $\geq25$ kg/m$^2$), smoking (never, former, current smoker), alcohol (never, former, current drinker), frequency of physical activity (0–2, 3–5, or 6-7/week), and level of high-density lipoprotein (HDL) ($<40$, $\geq40$ mg/dL in males, $<50$, $\geq50$ mg/dL in females). Spline plots to assess the non-linearity of exposure (SBP/DBP) were created to visualize the effects of SBP and DBP on all-cause and CVD mortality. Statistical significance was assumed at $P < 0.05$. All statistical analyses were performed using SAS Windows version 9.4 (Cary, North Carolina, USA).

## Results

### Demographics

In total, 14,375 participants were included in this study, including 5,712 males (39.7%) and 8,663 females (60.1%). The mean age of the participants was 53±15 years (S2 Table). During a mean follow-up of 13 years with 198,556 person-years, 2,457 participants died. Of these, 570 (23.2%),150 (6.1%), 268 (10.9%), 56 (2.3%), 743 (30.2%), and 247 (10.1%) cases were from CVDs, IHD, stroke, hypertensive disease, all types of cancer, and non-disease, respectively. Among 150 cases of death from IHD, 87 (58.0%) were due to AMI. A total of 268 deaths from stroke were composed of 80 (29.9%), 72 (26.9%), and 116 (43.3%) cases due to cerebral hemorrhage, cerebral infarction, and other strokes, respectively.

### Risk of CVD mortality in the high BP group relative to the low BP group

In the SBP group, the risk of CVD mortality showed a tendency to increase from the cut-off point of 120 mmHg (Figs 1 and 2). The risk of CVD mortality remained similar to the cut-off value of 126 mmHg. Compared with the SBP group, the risks of CVD mortality were shown to be greater in the DBP group. The risks of CVD mortality showed dramatic increases at the two main cut-off points of 90 and 100 mmHg.

### Risk of CVD mortality according to increased systolic BP and diastolic BP and combined analysis of systolic and/or diastolic hypertension

The association between SBP and DBP on specific categories for the risk of death from all-cause, CVD, cancer, and specific diseases were analyzed (Fig 3). The cut-off values for significantly increasing mortality were primarily evaluated for SBP and DBP, respectively (S3 and S4 Tables). Based on the primary results, hypertension was defined as systolic and diastolic readings of $\geq130/85$ mmHg, and four combined BP groups were established, followed by the current definition as follows: normal combined group (130>SBP, 85>DBP), high SBP-combined group (SBP $\geq130$, 85>DBP), high DBP-combined group (SBP $\geq 130$, DBP$\geq85$), and highest combined group (SBP$\geq130$, DBP$\geq85$).

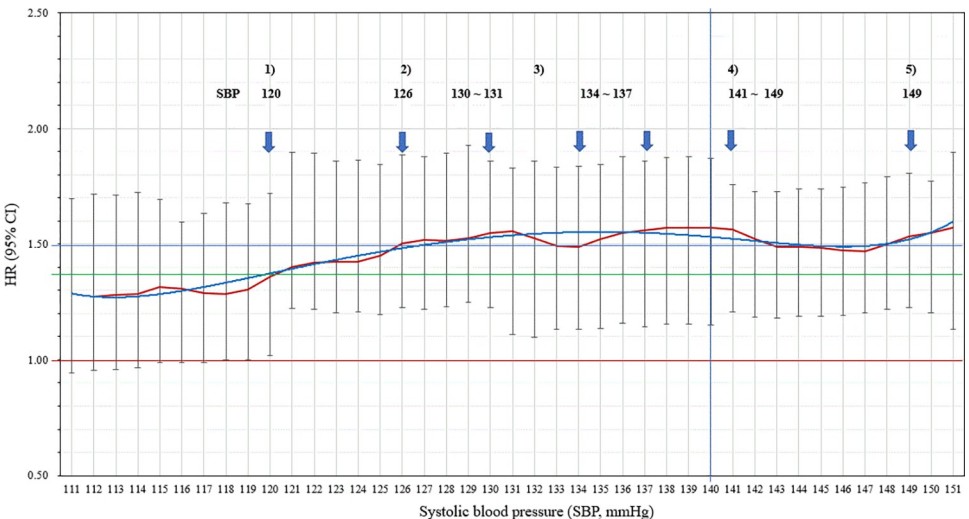

**Fig 1. The risk trend of cardiovascular mortality according to each specific systolic blood pressure (SBP) level.**

The association between the combined groups on specific categories and the risk of death is shown in Table 1. The risk of death from total CVD significantly increased in the high DBP-combined group (HR 1.34, 95% CI 1.26–2.00) and highest combined group (HR 1.84, 95% CI 1.51–2.25) (P trend <0.001). Among specific categories, there was a significantly increased risk of death from total stroke and hemorrhagic stroke in the three high BP combined groups compared with the normal combined group (all P trends <0.001). The risk of death from total stroke and hemorrhagic stroke was shown to be higher in the DBP-combined group (HR for stroke: 2.34, 95% CI 1.07–5.11; HR for hemorrhagic stroke: 4.11, 95% CI 1.40–12.06) than in the SBP-combined group (HR for stroke: 1.77, 95% CI 1.24–2.53; HR for hemorrhagic stroke: 1.97, 95% CI 1.05–3.68) (all P trends <0.001). In the highest combined group, the risk of death from all types of stroke was increased. There was no significant difference in the risk of death from IHD or AMI between the combined BP groups (P trend = 0.13 and 0.90, respectively).

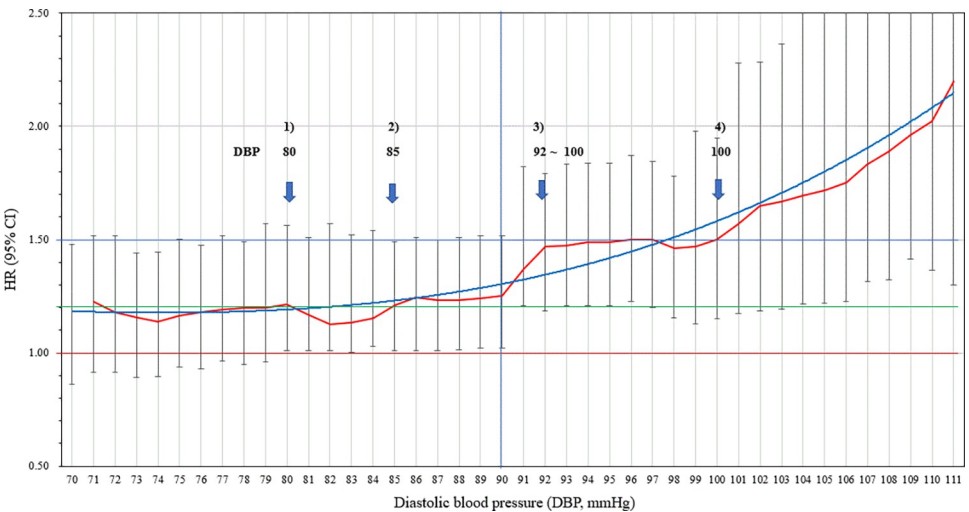

**Fig 2. The risk trend of cardiovascular mortality according to each specific diastolic blood pressure (DBP) level.**

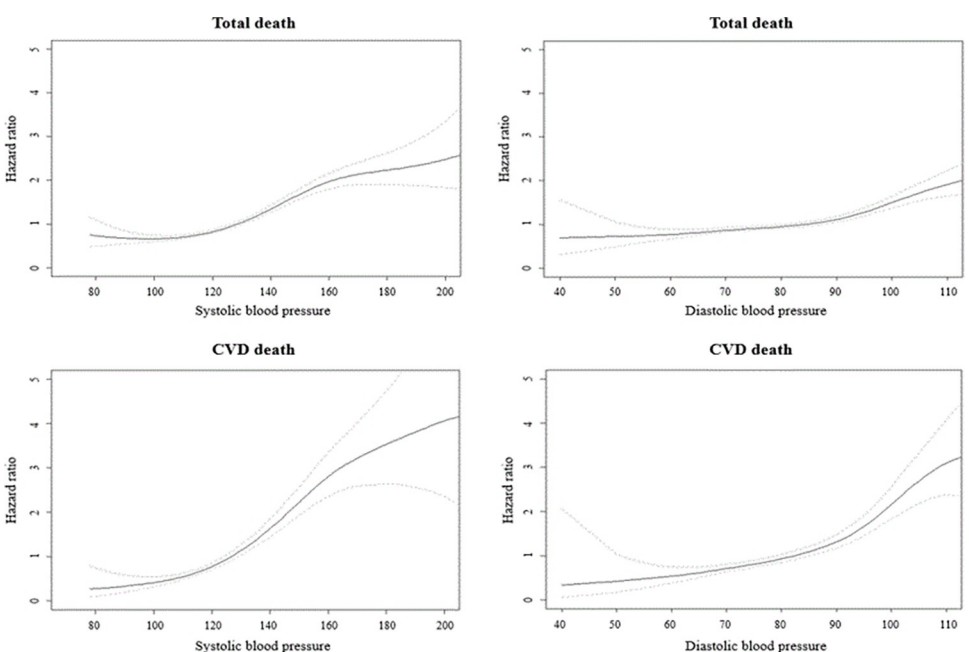

**Fig 3. Spline graph to visualize the effect of systolic blood pressure and diastolic blood pressure on the risk of death from all-cause and CVD.** (Abbreviation: CVD, cardiovascular disease).

## Combined effect of BP and other modifiers on CVD mortality: Ascertainment of the high risk group

In the stratification analysis, there was a significantly increased risk of death from total CVD in the highest combined BP group with a younger age (HR 3.51, 95% CI 2.52–4.69), anemia (HR 3.50, 95% CI 2.69–4.70), and high cholesterol (HR 2.86, 95% CI 1.98–4.14) compared with the normal group (all p <0.05). The risk of earlier death significantly increased (HR 11.50, 95% CI 3.90–32.03) in the highest combined BP group (p = 0.004). There was no significant trend in the risk of CVD mortality between the combined BP groups according to BMI, CKD, and DM (all p >0.05). Among specific CVD categories, the risk of death from stroke in the highest combined BP group significantly increased along with younger age (HR 6.01, 95% CI 3.25–11.47), anemia (HR 3.92, 95% CI 2.49–6.63), and high BMI (HR 9.72, 95% CI 3.47–27.25) compared with the normal group (all p <0.05). The risk of earlier death significantly increased (HR 18.99, 95% CI 4.56–84.60) in the highest combined BP group (p = 0.012). More detailed information is presented in Table 2.

## Discussion

As is widely known, the effect of high SBP is usually highlighted as a predictor of CVD events and related mortality [14,15]. However, only a few studies have begun to elucidate the significant effect of DBP on CVD morbidity and mortality [16,17]. Nevertheless, the combined effect of systolic and diastolic hypertension remains unclear. In the current study, we estimated the combined effect of systolic and/or diastolic hypertension and its combined effect with several modifiers on death. We found a higher risk of CVD-specific death as well as all-cause death in the highest combined group than in the isolated high SBP or high DBP group. This is particularly pronounced with regard to stroke-specific deaths. We also observed a significantly higher risk of CVD-specific death when participants had high BP with younger age, anemia, and high

**Table 1. Combined association of systolic blood pressure (SBP) and diastolic blood pressure (DBP) on the risk of mortality in the Korean Multi-center Cancer Cohort study over 15 follow-up years.**

|  | Death | SBP < 130 & DBP < 85 | SBP < 130 & DBP ≥ 85 | SBP ≥ 130 & DBP < 85 | SBP ≥ 130 & DBP ≥ 85 |  |
|---|---|---|---|---|---|---|
|  | N | HR | HR (95% CI) [1] | HR (95% CI) [1] | HR (95% CI) [1] | P-trend |
| All-cause | 2457 | 1.00 | 0.88 (0.64–1.21) | **1.15 (1.03–1.28)** | **1.20 (1.10–1.32)** | 0.004 |
| CVD | 570 | 1.00 | 1.70 (0.94–3.06) | **1.59 (1.26–2.00)** | **1.84 (1.51–2.25)** | <0.001 |
| IHD | 150 | 1.00 | 0.48 (0.07–3.47) | 1.15 (0.73–1.80) | 1.34 (0.92–1.95) | 0.13 |
| AMI | 87 | 1.00 | 0.62 (0.08–4.52) | 0.96 (0.54–1.72) | 1.04 (0.63–1.69) | 0.90 |
| Stroke | 268 | 1.00 | **2.34 (1.07–5.11)** | **1.77 (1.24–2.53)** | **2.59 (1.92–3.50)** | <0.001 |
| Hemorrhagic stroke | 80 | 1.00 | **4.11 (1.40–12.06)** | **1.97 (1.05–3.68)** | **2.29 (1.32–3.95)** | <0.001 |
| Ischemic stroke | 72 | 1.00 | 2.09 (0.49–8.97) | 0.98 (0.48–1.99) | **1.69 (1.04–3.08)** | 0.045 |
| Stroke, others | 116 | 1.00 | 0.95 (0.13–7.05) | **2.45 (1.39–4.30)** | **3.68 (2.26–5.98)** | <0.001 |
| Hypertension | 56 | 1.00 | 3.53 (0.80–15.62) | 1.91 (0.95–3.83) | 1.38 (0.71–2.69) | 0.29 |
| Cancer | 743 | 1.00 | 0.58 (0.31–1.09) | 1.01 (0.83–1.23) | 1.04 (0.87–1.23) | 0.65 |
| Non-disease | 247 | 1.00 | 0.44 (0.14–1.38) | 1.02 (0.72–1.44) | 0.94 (0.70–1.26) | 0.76 |
|  | Death | SBP < 140 & DBP < 90 | SBP < 140 & DBP ≥ 90 | SBP ≥ 140 & DBP < 90 | SBP ≥ 140 & DBP ≥ 90 |  |
|  | N | HR | HR (95% CI) [1] | HR (95% CI) [1] | HR (95% CI) [1] | P-trend |
| All-cause | 2457 | 1.00 | 0.93 (0.80–1.09) | 1.08 (0.94–1.24) | **1.25 (1.13–1.37)** | **<0.01** |
| CVD | 570 | 1.00 | 1.09 (0.79–1.51) | 1.22 (0.91–1.62) | **1.74 (1.44–2.10)** | **<0.01** |
| IHD | 150 | 1.00 | 1.11 (0.62–1.99) | 0.84 (0.45–1.55) | 1.32 (0.91–1.91) | 0.22 |
| AMI | 87 | 1.00 | 1.03 (0.49–2.20) | 0.54 (0.21–1.37) | 1.06 (0.65–1.75) | 0.93 |
| Stroke | 268 | 1.00 | 1.25 (0.77–2.04) | **1.69 (1.13–2.52)** | **2.39 (1.82–3.15)** | **<0.01** |
| Hemorrhagic stroke | 80 | 1.00 | 1.57 (0.73–3.38) | 1.23 (0.54–2.79) | **1.89 (1.14–3.15)** | **0.02** |
| Ischemic stroke | 72 | 1.00 | 1.82 (0.83–3.98) | 1.77 (0.86–3.63) | 1.52 (0.87–2.66) | 0.10 |
| Stroke, others | 116 | 1.00 | 0.51 (0.16–1.64) | **2.03 (1.11–3.72)** | **3.63 (2.39–5.52)** | **<0.01** |
| Hypertension | 56 | 1.00 | 0.67 (0.20–2.24) | 1.02 (0.42–2.49) | 1.23 (0.67–2.25) | 0.52 |
| Cancer | 743 | 1.00 | 0.79 (0.59–1.05) | 1.02 (0.79–1.33) | 1.10 (0.92–1.32) | 0.31 |
| Non-disease | 247 | 1.00 | 0.75 (0.46–1.22) | 0.74 (0.44–1.25) | 1.02 (0.75–1.39) | 0.87 |

Abbreviation: CVD, Cardiovascular diseases; CKD, Chronic kidney disease; DM, Diabetes mellitus

1. Adjusted for age, sex, past medical history of diabetes mellitus, family history of cardiovascular disease, BMI, cigarette smoking, alcohol consumption, physical activity, level of high-density lipoprotein by using a Cox proportional hazards regression analysis.

cholesterol levels. This combined effect is noticeable in the combined group with high SBP and DBP. Despite several previous studies evaluating the relationship between BP and CVD morbidity and mortality using national health checkup data from the Korean population [9,18–21], limited information on medication history, compliance, or other clinical risk factors remained as potential confounders. Thus, in this study, we sensitively excluded individuals with a previous history of antihypertensive medication or important comorbidities using a questionnaire collected by person-to-person interviews to improve the accuracy of our data.

Notably, we found interesting patterns of CVD mortality according to changes in each cut-off point in the SBP and DBP groups. The American College of Cardiology/American Heart Association (ACC/AHA) recently suggested lowering the margin of stage 1 hypertension to SBP of 130 mmHg-139 mmHg and DBP of 80 mmHg-89 mmHg [22]. The evidence used for this change to the treatment guidelines was mostly based on studies from Western populations, from which further studies are required. In the Asian population, we found a prior increase in the risk of CVD mortality before SBP of 130 mmHg and a similar or rather decreased risk of DBP of 80 mmHg-89 mmHg. We noticed several increases in the risk of

**Table 2. Combined association of systolic blood pressure (SBP) and diastolic blood pressure (DBP) on the risk of CVD mortality in stratified analysis by CVD risk factors.**

| | Death | SBP < 130 & DBP < 85 | SBP < 130 & DBP ≥ 85 | SBP ≥ 130 & DBP < 85 | SBP ≥ 130 & DBP ≥ 85 |
|---|---|---|---|---|---|
| | N | HR | HR (95% CI)[1] | HR (95% CI) [1] | HR (95% CI) [1] |
| Age at death [3] | Early death[5] | 1.00 | NA | NA | 11.50 (3.90–32.03) |
| | Common death [5] | 1.00 | 1.36 (0.76–2.46) | 2.45 (1.95–3.09) | 2.48 (1.82–3.35) |
| Age at enrollment [2] | Low age[5] | 1.00 | 0.43 (0.06–3.14) | 1.55 (0.86–2.80) | 3.51 (2.52–4.69) |
| | High age [5] | 1.00 | 2.11 (1.13–3.93) | 1.92 (1.49–2.48) | 1.96 (1.46–2.67) |
| Anemia [4] | No [5] | 1.00 | 1.80 (0.91–3.56) | 1.70 (1.29–2.25) | 1.37 (1.10–1.81) |
| | Yes [5] | 1.00 | 2.86 (0.88–9.29) | 1.52 (0.95–2.42) | 3.50 (2.69–4.70) |
| High cholesterol [4] | No [5] | 1.00 | 0.92 (0.50–2.35) | 1.49 (1.12–1.97) | 1.49 (1.17–1.91) |
| | Yes[5] | 1.00 | 1.39 (0.43–4.53) | 2.00 (1.31–3.06) | 2.86 (1.98–4.14) |
| Obesity [4] | BMI < 25 | 1.00 | 1.80 (0.88–3.68) | 1.59 (1.20–2.10) | 1.75 (1.37–2.24) |
| | BMI ≥ 25 | 1.00 | 2.15 (0.75–6.17) | 1.93 (1.17–3.19) | 2.27 (1.47–3.51) |
| CKD [4] | No | 1.00 | 0.96 (0.30–3.04) | 1.40 (1.00–1.97) | 1.55 (1.16–2.07) |
| | Yes | 1.00 | 1.66 (0.51–5.42) | 1.50 (0.96–2.35) | 1.56 (1.06–2.31) |
| DM [4] | No | 1.00 | 1.71 (0.87–3.36) | 1.65 (1.28–2.12) | 1.87 (1.50–2.33) |
| | Yes | 1.00 | 1.59 (0.48–5.26) | 1.20 (0.68–2.14) | 1.51 (0.94–2.41) |

Abbreviation: CVD, Cardiovascular diseases; CKD, Chronic kidney disease; DM, Diabetes mellitus; NA, Not available due to zero events.

1. Adjusted for age, sex, diabetes mellitus, cigarette smoking, alcohol consumption, physical activity, and family history of CVDs.

2. [Low age] included men of age < 55 years and women of age < 65; [High age] included men of age ≥ 55 years and women of age ≥ 65 years, respectively.

3. [Early death] was defined as death under the age of 55 years for men and under 65 years for women; whereas [Common death] was defined as death at age after the age of [Early death] for both men and women.

4. The criteria of anemia, high cholesterol, CKD, and DM at baseline were defined as follows: 'Hemoglobin levels ≤ 13 mg/dL for men and ≤ 12 mg/dL for women'; 'Total cholesterol levels of ≥ 200 mg/dL'; 'BMI ≥ 25 kg/m² under Asia-Pacific criteria'; 'estimated glomerular filtration rate (eGFR) < 60 ml/min/1.73 m²; and 'fasting blood sugar (FBS) levels ≥ 126 mg/dL or past history of DM diagnosis and anti-DM mediations', respectively.

5. Significant p-value for interaction: Age at death, p = 0.004; Age at enrollment, p = 0.048; Anemia, p = 0.03; High cholesterol levels, p = 0.006.

CVD mortality every 10 mmHg in DBP between 90 mmHg and 110 mmHg. This result can be explained by the difference in hemodynamic effects of arterial stiffness on BP elevation between ethnicities [23]. Based on the fact that arterial stiffness contributes to vascular structure remodeling and is considered a significant predictor of future CVD events and death [24,25], it became important to clarify the potential disparities in ethnic-specific risks of elevated BP and related vascular diseases. Despite a lack of data on arterial stiffness in the Asian population, a previous study reported higher central arterial stiffness with an elevated risk for CVD in South Asians compared to Caucasians [26]. With more susceptibility to BP changes resulting from higher arterial stiffness, there might be a sensitive response that leads to greater BP changes, particularly in DBP, in Asian populations.

This explanation can also be applied to our results. Among the disease-specific mortalities, deaths from stroke had a significantly higher risk in the highest combined BP group. Since arterial stiffness is significantly correlated with intracranial large artery disease among ethnic Chinese and South Asian populations [27], we found a consistent result of deaths from stroke with a significantly higher risk in this study. In addition, elevated DBP has been reported to play a sensitive role in death from hemorrhagic stroke [28]. Several studies have previously emphasized the association between high DBP and ischemic or hemorrhagic stroke [29–31]. Based on our results, we also demonstrated the importance of DBP superimposed on high SBP on CVD mortality. Therefore, we could support the new recommendation for early intervention of BP and management of DBP in the Asian population to prevent CVD mortality.

Finally, participants with high BP were found to have a high risk for total CVD and stroke-specific mortality. The combined effect of high combined BP and anemia played a significant role in increasing the mortality of both CVD and stroke. Based on the fact that the presence of anemia is an independent risk factor for CVD events [32–34], anemic status accompanied by high BP might exacerbate the ischemic condition, resulting in higher CVD morbidity and mortality. Contrary to our expectation, high combined BP was associated with increased total CVD and stroke-specific mortality in the younger age group, as well as the risk of earlier death. This might reflect the genetic susceptibility of early vascular aging, which has a steep progression and could be a strong predictor of CVD outcomes [35]. Future studies are needed to verify this.

This study has several limitations. First, BP values were measured once at enrollment. Therefore, it is unclear whether the participants with high BP had permanently elevated BP. However, it has been reported that the effects of additional BP measurements did not change the risk evaluated by a single measurement [36]. In addition, based on the result of a previous study reporting increased incidence rates for future hypertension in prehypertension groups [37,38], we do not believe this limitation would have significantly changed the main outcome. Second, follow-up on possible antihypertensive treatment after enrollment was not performed. Thus, we could not consider the effects of such treatments on CVD mortality. Well-controlled studies, including information on antihypertensive treatment, are required to verify this.

However, the data used in this study have several strengths. We used well-refined, reliable, community-based, prospective cohort data. Anthropometric indices were collected using standard methods during the physical examination, and lifestyle factors were recorded using direct interviews with structured questionnaires by well-trained interviewers. This meant that we could select more homogenous study participants than those of previous studies. In addition, we obtained highly reliable mortality data from the national death certificate system for analysis [39].

In conclusion, we demonstrated the detailed role of combined BP, the sensitive role of DBP, and the pattern of CVD mortality according to binary cut-off points. Based on our results, combined BP could be a predictor of cardiovascular death as opposed to an independent predictor. In addition, we suggest that previously neglected prehypertensive groups also have an increased risk for CVD mortality, suggesting that intervention with antihypertensive medication could be effective. Our results may help clinicians to better manage and provide information to patients with a high risk of CVD mortality.

## Supporting information

**S1 Fig. Identifying process of study group in Korean Multi-center Cancer Cohort (KMCC) study database.** Overview of the study participants from the KMCC study used to analyze the association between BP and cardiovascular death (Abbreviation: SBP, systolic blood pressure; DBP, diastolic blood pressure).
(TIF)

**S1 Table. Classification of the cause of death based on the ICD-10.**
(DOCX)

**S2 Table. General characteristics of study participants by systolic blood pressure and diastolic blood pressure of 14,375 cohort participants in the Korean Multi-center Cancer Cohort study (KMCC) over 15 follow-up years.**
(DOCX)

**S3 Table. The association with systolic blood pressure on different specific categories for the risk of mortality in the Korean Multi-center Cancer Cohort study over 15 follow-up years.**
(DOCX)

**S4 Table. The association with diastolic blood pressure on different specific categories for the risk of mortality in the Korean Multi-center Cancer Cohort study over 15 follow-up years.**
(DOCX)

**S1 File. Sample data of Korean Multi-center Cancer Cohort (KMCC) during 1993 to2004 including a few essential variables in the study.**
(XLSX)

## Author Contributions

**Conceptualization:** Seokyung An, Aesun Shin, Sue K. Park.

**Data curation:** Sangjun Lee, Seokyung An, Aesun Shin, Sue K. Park.

**Formal analysis:** Sangjun Lee, Seokyung An, Aesun Shin, Sue K. Park.

**Funding acquisition:** Sue K. Park.

**Investigation:** Ji Hoon Hong, Sangjun Lee, Sue K. Park.

**Methodology:** Sangjun Lee, Aesun Shin, Sue K. Park.

**Project administration:** Aesun Shin, Sue K. Park.

**Supervision:** Ju-Yeun Lee, Ji Hoon Hong, Seokyung An, Aesun Shin, Sue K. Park.

**Validation:** Ju-Yeun Lee, Ji Hoon Hong, Sangjun Lee, Sue K. Park.

**Writing – original draft:** Ju-Yeun Lee.

**Writing – review & editing:** Ju-Yeun Lee, Sue K. Park.

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
