## [Decision Letter · Decision Letter 0]

16 May 2022

PONE-D-22-06890Binary cutpoint and the combined effect of systolic and diastolic blood pressure on cardiovascular disease mortality: a community-based cohort studyPLOS ONE

Dear Dr. Park,

Thank you for submitting your manuscript to PLOS ONE. After careful consideration, we feel that it has merit but does not fully meet PLOS ONE’s publication criteria as it currently stands. Therefore, we invite you to submit a revised version of the manuscript that addresses the points raised during the review process.

We look forward to receiving your revised manuscript.

Kind regards,

Tariq Jamal Siddiqi

Academic Editor

PLOS ONE

Journal Requirements:

a) Did participants provide their written or verbal informed consent to participate in this study?

Reviewers' comments:

Reviewer's Responses to Questions

**Comments to the Author**

1. Is the manuscript technically sound, and do the data support the conclusions?

Reviewer #1: Yes

2. Has the statistical analysis been performed appropriately and rigorously? 

Reviewer #1: Yes

3. Have the authors made all data underlying the findings in their manuscript fully available?

Reviewer #1: Yes

4. Is the manuscript presented in an intelligible fashion and written in standard English?

Reviewer #1: Yes

5. Review Comments to the Author

Reviewer #1: The authors presented a manuscript evaluating the combined effect of systolic and diastolic blood pressure on cardiovascular disease mortality. The analysis is well done and the manuscript has been drafted in an intelligible manner. Figures and tables are presented appropriately. However, there are a few issues which need to be addressed:

1. The authors should consider adding numbers and percentages to provide a comprehensive view of the mortality and morbidity burden in the introduction.

2. Kindly mention the full form of Fig as figure, and report table citations as Table S2 instead of S2 table.

3. The authors should report the p values associated with results mentioned within the text of the manuscript.

4. Kindly ensure that all HRs are reported along with their 95% confidence intervals.

6. PLOS authors have the option to publish the peer review history of their article (what does this mean?). If published, this will include your full peer review and any attached files.

Reviewer #1: No

---

## [Author Response · Author response to Decision Letter 0]

10 Jun 2022

May 19, 2022

PONE-D-22-06890

Binary cutpoint and the combined effect of systolic and diastolic blood pressure on cardiovascular disease mortality: a community-based cohort study

Dear editor and reviewers

We appreciate reviewers’ constructive comments on our manuscript. We have carefully considered these, and enclose a revised version of the manuscript which incorporates the reviewer’s comments. Our point-by-point responses to the comments are listed below. If you have any question, please feel free to contact us. We look forward to a favorable decision from the Journal. Thank you for your time for this revised manuscript.

Sincerely, 

Sue K Park, MD, PhD

Review Comments to the Author

Reviewer #1: The authors presented a manuscript evaluating the combined effect of systolic and diastolic blood pressure on cardiovascular disease mortality. The analysis is well done and the manuscript has been drafted in an intelligible manner. Figures and tables are presented appropriately. However, there are a few issues which need to be addressed:

Authors’ response: We appreciate reviewer’s comment. We carefully addressed all issues and our point-by-point responses to the comments are listed below. 

1. The authors should consider adding numbers and percentages to provide a comprehensive view of the mortality and morbidity burden in the introduction.

Authors’ response: We appreciate reviewer’s insightful comment. Numbers and percentages to provide a comprehensive view of the mortality and morbidity burden has been added in the introduction section per reviewer’s opinion. 

In the revised manuscript (p3. Lines 48-52)>

It was reported that county‐level hypertension‐related CVD mortality increased from 362.1 per 100,000 in 2000 to 430.1 per 100,000 in 2019 among adults aged ≥65 years. Elevated BP-related CVD mortality during 2010 to 2019 was found to increase 86.2% among patients aged 35 to 64 years, and 66.1% for patients aged ≥65 years.

2. Kindly mention the full form of Fig as figure, and report table citations as Table S2 instead of S2 table.

Authors’ response: We appreciate reviewer’s helpful comment. We followed the PLOS ONE's style requirements including those for file naming (such as Fig 1 and S1 Table). We appreciate reviewer for giving us the opportunity to check the journal style again. For convenience, the journal style requirements are posted at the bottom. If reviewer strongly wants to change the naming, we are willing to consult with the editor's office. 

3. The authors should report the p values associated with results mentioned within the text of the manuscript.

Authors’ response: We appreciate reviewer’s helpful comment. We provided p values for all results mentioned within the text per reviewer’s opinion.

In the revised manuscript (p10. lines 187-192)>

The risk of death from total stroke and hemorrhagic stroke was shown to be higher in the DBP-combined group (HR for stroke: 2.34, 95% CI 1.07-5.11; HR for hemorrhagic stroke: 4.11, 95% CI 1.40-12.06) than in the SBP-combined group (HR for stroke: 1.77, 95% CI 1.24-2.53; HR for hemorrhagic stroke: 1.97, 95% CI 1.05-3.68) (P trends <0.001). In the highest combined group, the risk of death from all types of stroke was increased. There was no significant difference in the risk of death from IHD or AMI between the combined BP groups (P trend =0.13 and 0.90, respectively).

4. Kindly ensure that all HRs are reported along with their 95% confidence intervals.

Authors’ response: We appreciate reviewer’s helpful comment. We added 95% CI to all HRs in the results section. 

In the revised manuscript (p10. lines 187-192)>

The risk of death from total stroke and hemorrhagic stroke was shown to be higher in the DBP-combined group (HR for stroke: 2.34, 95% CI 1.07-5.11; HR for hemorrhagic stroke: 4.11, 95% CI 1.40-12.06) than in the SBP-combined group (HR for stroke: 1.77, 95% CI 1.24-2.53; HR for hemorrhagic stroke: 1.97, 95% CI 1.05-3.68). In the highest combined group, the risk of death from all types of stroke was increased. There was no significant difference in the risk of death from IHD or AMI between the combined BP groups.

---

## [Editor Report · Decision Letter 1]

12 Jun 2022

Binary cutpoint and the combined effect of systolic and diastolic blood pressure on cardiovascular disease mortality: a community-based cohort study

PONE-D-22-06890R1

Dear Dr. Park,

We’re pleased to inform you that your manuscript has been judged scientifically suitable for publication and will be formally accepted for publication once it meets all outstanding technical requirements.

Kind regards,

Tariq Jamal Siddiqi

Academic Editor

PLOS ONE
---

## [Editor Report · Acceptance letter]

23 Jun 2022

PONE-D-22-06890R1 

Binary cutpoint and the combined effect of systolic and diastolic blood pressure on cardiovascular disease mortality: a community-based cohort study 

Dear Dr. Park:

I'm pleased to inform you that your manuscript has been deemed suitable for publication in PLOS ONE. Congratulations! Your manuscript is now with our production department. 

Kind regards, 

on behalf of

Dr. Tariq Jamal Siddiqi 

Academic Editor

PLOS ONE